# Transgenerational Transmission of 2,3,7,8-Tetrachlorodibenzo-p-dioxin (TCDD) Effects in Human Granulosa Cells: The Role of MicroRNAs

**DOI:** 10.3390/ijms25021144

**Published:** 2024-01-17

**Authors:** Laura Gaspari, Delphine Haouzi, Aurélie Gennetier, Gaby Granes, Alexandra Soler, Charles Sultan, Françoise Paris, Samir Hamamah

**Affiliations:** 1Unité d’Endocrinologie-Gynécologie Pédiatrique, Service de Pédiatrie, Hôpital Arnaud-de-Villeneuve, CHU Montpellier, Université de Montpellier, 34295 Montpellier, France; dr.lauragaspari@gmail.com (L.G.); pr.charles.sultan@gmail.com (C.S.);; 2Centre de Référence Maladies Rares du Développement Génital, Constitutif Sud, Hôpital Lapeyronie, CHU Montpellier, Université de Montpellier, 34295 Montpellier, France; 3INSERM U 1203, Développement Embryonnaire Fertilité Environnement, Université de Montpellier, INSERM, 34295 Montpellier, Francesoler.alexandra@live.fr (A.S.); 4Département de Biologie de la Reproduction et DPI (ART/PGD), Hôpital A. de Villeneuve, CHU Montpellier, Université de Montpellier, 34295 Montpellier, France; 5Global ART Innovation Network (GAIN), 34295 Montpellier, France

**Keywords:** epigenetics, miRNA, transgenerational transmission, dioxin, TCDD, endocrine-disrupting chemicals (EDCs), granulosa cells, cancers

## Abstract

Endocrine-disrupting chemicals (EDCs) might contribute to the increase in female-specific cancers in Western countries. 2,3,7,8-tetrachlordibenzo-p-dioxin (TCDD) is considered the “prototypical toxicant” to study EDCs’ effects on reproductive health. Epigenetic regulation by small noncoding RNAs (sncRNAs), such as microRNAs (miRNA), is crucial for controlling cancer development. The aim of this study was to analyze transcriptional activity and sncRNA expression changes in the KGN cell line after acute (3 h) and chronic (72 h) exposure to 10 nM TCDD in order to determine whether sncRNAs’ deregulation may contribute to transmitting TCDD effects to the subsequent cell generations (day 9 and day 14 after chronic exposure). Using Affymetrix GeneChip miRNA 4.0 arrays, 109 sncRNAs were found to be differentially expressed (fold change < −2 or >2; *p*-value < 0.05) between cells exposed or not (control) to TCDD for 3 h and 72 h and on day 9 and day 14 after chronic exposure. Ingenuity Pathway Analysis predicted that following the acute and chronic exposure of KGN cells, sncRNAs linked to cellular development, growth and proliferation were downregulated, and those linked to cancer promotion were upregulated on day 9 and day 14. These results indicated that TCDD-induced sncRNA dysregulation may have transgenerational cancer-promoting effects.

## 1. Introduction

The rapid increase in female-specific cancers in the Western world over the last 50 years has led researchers to suggest the contribution of environmental factors, especially endocrine-disrupting chemicals (EDCs) [1,2]. EDCs’ oncogenic effects in women have been confirmed by experimental studies using different molecules, including pesticides, dithiothreitol, phthalates, bisphenol A, diethylstilbestrol (DES), and dioxin heavy metals [1,3]. Exposure to estrogenic EDCs during fetal development is particularly harmful because they can alter the hormonal environment, thus predisposing the fetus to genital abnormalities, infertility, and reproductive tract tumors later in life [4].

Among the expanding list of EDCs, 2,3,7,8-tetrachlordibenzo-p-dioxin (TCDD, or dioxin) is considered the most toxic chemical produced by humans [5] and has been selected as the “prototypical toxicant” for studying EDCs’ transgenerational effects on reproductive health [6]. TCDD is a ubiquitous contaminant introduced in the environment as an unwanted by-product of manufacturing, combustion, waste incineration, car traffic, and cigarette smoking as well as forest fires and volcanic eruptions [6,7,8]. Much of our knowledge on TCDD exposure in humans comes from two sources: exposure to Agent Orange in South Vietnam, Laos, and Cambodia during the Second Indochina War (from 1962 to 1971) and an industrial accident in Seveso, Italy, in 1976 [9,10,11]. Due to TCDD’s solubility in lipids, high chemical stability, and resistance to biodegradation, exposure resulted in severe and long-term effects in women and their offspring [12,13,14,15,16,17,18].

Although the factors involved in TCDD-induced toxicity are still under investigation, several studies showed that the female endocrine system-disrupting effects of TCDD are, at least in part, caused by direct action on the ovaries [19,20]. Aromatic hydrocarbon receptors (AhR), the main mediator of TCDD intracellular action, are expressed in mammalian ovarian tissue, including human granulosa cells [19,20,21,22,23]. In porcine ovarian granulosa cells, TCDD alters the expression of genes involved in cell cycle regulation, proliferation, and follicular atresia [23,24,25,26,27,28] and it also modifies the expression of proteins involved in the rearrangement of the cytoskeleton and extracellular matrix and in the cellular response to stress [29]. In addition, many genes involved in cell proliferation and differentiation, inflammation, stress response, apoptosis, and oncogenesis are differentially expressed in AhR knock-down granulosa cells incubated with TCDD compared with non-exposed cells. This suggests that TCDD may affect its target tissues also in an AhR-independent manner [23].

TCDD is considered a non-genotoxic carcinogen (also termed epigenetic or non-DNA reactive carcinogen) because it does not appear to cause direct genetic damage [30], but it induces a state of genomic instability (GI) in which genetic damage is observed only many cell generations later in the progeny of the exposed cells [7,31]. As no immediate genotoxicity is observed, an epigenetic mechanism of such delayed genetic damage could be hypothesized, possibly associated with changes in small noncoding RNAs’ (sncRNAs), such as microRNAs (miRNA), expression and DNA methylation patterns [7,11,31,32].

Therefore, the aim of this study was to analyze transcriptional activity and miRNA expression changes in human granulosa cells after acute and chronic exposure to TCDD in order to determine whether miRNA deregulation may be involved in the transgenerational effects of TCDD. Indeed, miRNAs are implicated in the interactive relationship of genomic landscape, gene environment, and disease phenotype [33].

As TCDD is considered an important paradigm to understand the health consequences of human exposure to EDCs, this study might contribute to explaining the role of the post-exposure miRNA deregulation in the increased incidence of female-specific cancers in the last 50 years.

## 2. Results

### 2.1. Choice of TCDD Concentration and Exposure Times

To investigate TCDD effects and to choose the lowest effective dose, we analyzed the expression of the genes encoding the aromatases *CYP1A1* and *CYP1B1*, which are induced by TCDD, in KGN cells exposed or not (vehicle alone, DMSO, and control) to different TCDD concentrations (1 nM, 10 nM, 50 nM, and 100 nM) and for different times. We chose this human ovarian granulosa-like tumor cell line because it is considered a human granulosa cell model (45). We observed a significant effect on *CYP1A1* expression, and to a lower extent also on *CYP1B1* expression, upon exposure to 10 nM TCDD for 3 h (15.5-fold and 6.7-fold increase compared to control, *p* < 0.001 for both) (Figure 1a,b). Therefore, for the subsequent experiments, we used 10 nM TCDD and 3 h and 72 h of exposure (i.e., acute and chronic exposure, respectively).

### 2.2. Characterization of the Morphology and Viability of Human Granulosa Cells

We did not detect any difference in KGN cell morphology between the control (DMSO) and exposed cells (1 nM, 10 nM, 50 nM, and 100 nM TCDD for 3 h, 6 h, 48 h, or 72 h). Figure 2 shows representative images of the control and treated KGN cells (10 nM TCDD for 72 h).

Cell proliferation (trypan blue viability assay) was similar in the control and treated KGN cells (10 nM TCDD for 3 h and 72 h and on day 9 and day 14 after chronic exposure) (Figure 3). KGN cell proliferation was not affected also after incubation with 1 nM, 50 nM, and 100 nM TCDD for up to 72 h (Appendix A).

### 2.3. CYP1A1 and CYP1B1 Expression after Acute and Chronic TCDD Exposure and on Day 9 and 14 after Chronic Exposure

We determined *CYP1A1* and *CYP1B1* expression levels after acute (3 h) and chronic (72 h) exposure to 10 nM TCDD or DMSO (control) and also on day 9 and day 14 after chronic exposure (Figure 4). TCDD acute and chronic exposure significantly increased *CYP1A1* expression by 11 (*p* < 0.001) and 7 times (*p* = 0.005) and *CYP1B1* expression by nearly 4 (*p* < 0.001) and 7 times (*p* < 0.001), respectively. Conversely, on day 9 and day 14 after chronic exposure, the expression levels of both genes were comparable in the control and exposed KGN cells (*p* > 0.1).

### 2.4. KGN Cell miRNome Profile in Function of the Exposure Status

In total, the microarray analysis of exposed (10 nM TCDD) and control (DMSO) KGN cells identified 109 sncRNAs that were differentially expressed (fold change < −2 or >2 and *p*-value < 0.05, ANOVA test): 21 sncRNAs after 3 h of exposure (RNA extracted after 48 h), 39 sncRNAs after 72 h of exposure, 18 sncRNAs after 9 days, and 31 sncRNAs after 14 days (Table 1).

Supervised hierarchical clustering on the basis of the expression of these 109 sncRNAs showed an adequate segregation between exposed and control KGD cells at each time-point (3 h, Figure 5; 48 h, Figure 6; day 9, Figure 7; day 14, Figure 8). Then, we selected seven differentially expressed miRNAs (miR-1231 upregulated upon TCDD exposure for 3 h; U49A and U49B upregulated upon TCDD exposure for 72 h; miR-24-2-5p and miR 30b-5p downregulated on day 9; and miR 27a-3p, miR 8063, U49A, and U49B downregulated on day 14 compared with control cells), using the Wilcoxon matched-pairs signed rank test, *t* test, or Mann–Whitney test, for validation by RT-qPCR. RT-qPCR confirmed that compared with the control (DMSO) KGD cells, U49A was upregulated after chronic (72 h) exposure to TCDD and downregulated on day 14 after chronic exposure (*p* < 0.001). Moreover, miR 24-2-5p and miR 30b-5p were downregulated on day 9 after chronic exposure to TCDD and miR 27a-3p on day 14 after chronic exposure (*p* < 0.001) (Figure 9). Conversely, we could not assess the expression of miR-1231 after acute exposure, U49B after chronic exposure, and miR 8063 and U49B on day 14 after chronic exposure because the cycle threshold (Ct) values were >40 cycles.

The Venn diagram (Figure 10) showed a minimal overlapping in deregulated sncRNAs among conditions, with the exception of day 14 after chronic exposure. Ingenuity Pathway Analysis (IPA) of the known and predicted targets of the sncRNAs differentially expressed in the function of the exposure status indicated that the sncRNAs deregulated upon acute (3 h) and chronic (72 h) TCDD exposure to targeted genes implicated in cellular development (1566 and 1774, respectively) and in cellular growth and proliferation (1472 and 1700, respectively) (Table 2). This analysis predicted that these target genes were downregulated as a result of their sncRNAs’ deregulation. The IPA results also suggested that day 9 after chronic TCDD exposure might be a crucial time because the genes predicted to be downregulated after acute and chronic exposure started to be expressed again (Table 2). Moreover, IPA predicted the upregulation of target genes linked to cell survival on day 14 (Table 2).

In addition, IPA indicated that genes targeted by these 109 sncRNAs were enriched in signaling pathways linked to cancer promotion (Figure 11). Specifically, IPA predicted that 20–30% of target genes included in the molecular mechanisms of cancer, tumor microenvironment pathway, ovarian cancer signaling, STAT3 pathway, WNT/β-catenin signaling, and oncostatin M signaling were downregulated after acute (3 h) and chronic (72 h) TCDD exposure and 20–27% of them were upregulated on day 14 after chronic exposure (Figure 11). The miRNA biogenesis signaling pathway showed the same pattern (Figure 12), suggesting a role of these sncRNAs in GI transmission in the progeny of TCDD-exposed cells.

Prediction of the miR-1231, miR-24-2-5p, miR 30b-5p, miR 27a-3p, and miR 8063 targets using four online databases (TargetScan7, miRTarBase, miRDB, and DIANA-TarBase v7.0) identified unique targets from the intersection of their results with the protein–protein interaction networks derived from Ingenuity (http://www.ingenuity.com, accessed on 18 September 2023). We considered only target genes with a context score percentile > 70 (Figure 13). To perform an indirect validation of the predicted miR-1231, miR-24-2-5p, miR 30b-5p, miR 27a-3p, and miR 8063 targets, we determined their expression after acute (3 h) and chronic (72 h) exposure to 10 nM TCDD or DMSO (control) and on day 9 and day 14 after chronic exposure (Figure 14). *MYC* and *FAS* expression levels were increased by almost 3–7 times (*p* ≤ 0.001) on day 9 and day 14 but not after chronic and acute exposure. B-cell lymphoma 2 (*BCL2*) and fibroblast growth factor (*FGF2*) expression levels also were increased by 5–6 times (*p* < 0.001) on day 9 and day 14, whereas they were significantly decreased after acute TCDD exposure (*p* = 0.002 and *p* = 0.08, respectively). Janus kinase (*JAK2*), vascular endothelial growth factor (*VEGF)* B and C, SMAD Family Member 5 (*SMAD5*), Forkhead box O1 (*FOXO1*), and collagen triple helix repeat containing-1 (*CTHRC1*) expression levels were increased by 3–7 times (*p* < 0.001) only on day 14 (Figure 14). The analysis of predicted targets for these miRNAs indicated potential interactions with genes involved in diverse cellular pathways, including signaling networks associated with cancer promotion, molecular mechanisms of cancer, and cellular transmission (Figure 15).

## 3. Discussion

Changes at the cellular, genetic, and epigenetic levels contribute to cancer development and progression [34]. Epigenetic regulations by sncRNAs, such as miRNAs, are crucial for controlling cancer pathogenesis by regulating gene transcription, translation, and activity [34]. Indeed, miRNAs can act as tumor suppressors or as oncogenes in the function of how they modulate their target genes. Here, by performing miRNA profiling in human granulosa cells after chronic TCDD exposure, we identified expression changes in miRNAs that target genes implicated in cancer-related signaling not only directly after acute or chronic exposure but also several days after the end of chronic exposure (i.e., in cells that were never exposed to TCDD). Several studies analyzed steroid hormone secretion, gene expression, the proteome, and the incidence of apoptosis in granulosa cells after TCDD exposure [5,7,8,20,23,25,27,28,29,35,36,37,38,39,40,41,42,43,44,45,46,47,48,49,50,51,52,53]. The present work was the first to assess the miRNome after acute and chronic TCDD exposure and in the following cell generations up to day 14 after chronic exposure. This analysis led to the identification of 109 sncRNAs that were differentially expressed between TCDD-exposed and control cells at one of the four time-points under study. The number of differentially expressed sncRNAs presented two peaks at 72 h of TCDD incubation (chronic exposure) and at day 14 after the chronic exposure’s end, suggesting a possible role of sncRNAs in the signaling pathways affected by EDC chronic exposure and in the subsequent cell generations [54,55,56].

Moreover, the IPA of these 109 sncRNAs clearly predicted a progressive downregulation of target genes linked to cell development, growth, proliferation, and survival after 3 h and 72 h of TCDD exposure. This is in agreement with the study by Heimler et al., who showed in cultured human granulosa cells that TCDD significantly increases apoptosis in a dose-dependent manner after 48 h of exposure [37]. Moreover, IPA predicted that the same target genes linked to cell development, growth, proliferation, and survival would be upregulated on day 14 after chronic exposure’s end (Figure 10), suggesting a possible transgenerational effect after the pivotal time of day 9. It has to be noted that TCDD- and DMSO-exposed cells displayed similar morphology (Figure 2) and proliferation (viability) (Figure 3) at all time-points under study, probably due to lower sensitivity of the hemocytometer analysis compared with the microscopic techniques used by Heimler et al. [37].

The main signaling pathways identified by IPA were linked to cancer promotion, also supporting our hypothesis of the transgenerational oncogenic effects of TCDD. Moreover, the miRNA biogenesis signaling pathway was downregulated at 72 h of TCDD exposure and strongly upregulated on day 14 after chronic exposure, suggesting a miRNA role in GI transmission and consequently in the higher cancer-promoting signaling in the progeny of TCDD-exposed cells, as previously reported in the ovaries of female mice chronically exposed to an EDC mixture [57].

Among the seven sncRNAs selected for validation, miR-1231 is a tumor suppressor miRNA. Using functional assays, Xie et al. showed that miR-1231 upregulation inhibits cell proliferation and cell cycle progression in ovarian cancer [58]. In KGN cells, miR-1231 was upregulated after acute TCDD exposure and IPA predicted a following downregulation of genes linked to cell development, growth, proliferation, and survival and also of transcripts included in the tumor signaling pathways. Moreover, *FGF2* and *BCL2* are direct miR-1231 targets. FGF2 is a pleiotropic cytokine that plays an essential role in damaged tissue repair and regeneration [59]. Previous studies found that *FGF2* downregulation in injured granulosa cells promotes their apoptosis by inhibiting autophagy [59]. Similarly, BCL-2 reduces apoptosis by inhibiting mitochondrial permeabilization, leading to the release of caspase activators and caspase-independent death effectors [60]. Yu et al. found that in cultured human umbilical vein endothelial cells, TCDD promotes apoptosis through activation of the EP3/p38MAPK/BCL-2 pathway and BCL-2 downregulation [61]. *FGF2* and *BCL2* downregulation after 3 h of TCDD exposure could be a direct effect of miR-1231 upregulation and one of the key players of apoptosis signaling (IPA analysis) in KGN cells after acute exposure.

MiR-24-2-5p and miR 30b-5p were downregulated on day 9 after chronic exposure. MiR-24 is generally considered a tumor-promoting miRNA, involved in cell proliferation, differentiation, invasion, and metastasis [62], but it can also have a tumor suppressor role in specific cancer types [63]. To our knowledge, this is the first time that miR-24 has been involved in ovarian cancer development. In KGN cells, miR-24-2-5p was downregulated on day 9 after chronic TCDD exposure. Moreover, the bioinformatic analysis showed that miR-24-2-5p was complementary to two genes: *MYC* and *SMAD2*. *MYC* is an oncogene frequently amplified in ovarian cancer and required for cancer cell growth [64]. MiR-24-2-5p downregulation was significantly correlated with the upregulation of *MYC* that remained upregulated also on day 14 after chronic exposure. Krüppel-like factors (KLFs) are a group of DNA-binding transcriptional regulators with essential functions in various cellular processes, including proliferation, migration, inflammation, and angiogenesis [65]. KLF expression is often altered in tumors, and this is essential for tumor development. For instance, *KLF8* is activated by transforming growth factor-β1 (TGF-β1) via *SMAD2* and contributes to ovarian cancer progression [65,66]. Cherukunnath et al. reported that the TGF-β1/SMAD2/KLF8 axis regulates epithelial–mesenchymal transition and contributes to ovarian cancer progression [66]. These authors also found that upon *KLF8* overexpression, ovarian cancer cells display increased *BCL2* expression that promotes cell survival [66]. In agreement, in KGN cells, *SMAD2* and *BCL2* were overexpressed on day 9 and also on day 14 after chronic exposure to TCDD (Figure 14).

In ovarian cancer cells, miR-30b-3p acts as a cancer suppressor [67]. Overexpression of miR-30b-3p in ovarian cancer cells inhibited cell migration and invasion by targeting *CTHRC1* that plays an important role in epithelial–mesenchymal transition [67]. Indeed, increased *CTHRC1* expression in epithelial ovarian cancer cells induces epithelial–mesenchymal transition, thereby promoting tumor cell invasion and metastasis [67,68]. In KGN cells, miR-30b-5p was downregulated and *CTHRC1* was overexpressed on day 9 after chronic TCDD exposure. *CTHRC1* was upregulated also on day 14 after chronic exposure, without miR-30b-5p downregulation. Many reports suggest that *CTHRC1* could exert different effects through several signaling pathways, such as TGF-β, WNT, integrin β/FAK, Src/FAK, MEK/ERK, PI3K/AKT/ERK, HIF-1α, and PKC-δ/ERK [69]. Therefore, its upregulation on day 14 after chronic TCDD exposure might be due to other miRNAs that modulate target genes in these signaling pathways.

MiR-27a-3p and miR-8063 were downregulated on day 14 after chronic exposure. Increasing evidence suggests that miR-27a is implicated in modulating tumorigenesis, proliferation, apoptosis, invasion, migration, and angiogenesis [70]. This miRNA is considered to be an oncogene in many tumor types, including prostate, liver, colorectal, and non-small cell lung cancer. However, some studies suggest that miR-27a functions as a tumor suppressor in small cell lung cancer and cervical adenocarcinoma [71]. Wang et al. found that miR-27a promotes ovarian cancer progression through its target *FOXO1* [71], while genistein, a nontoxic miR-27a inactivator, can block ovarian cancer cell growth and migration [72]. Conversely, Nie et al. found that in human granulosa cells, miR-27a targets *SMAD5*, but not *SMAD2* and *SMAD4*, and promotes apoptosis via the FAS ligand–FAS pathway [73]. In KGN cells, *SMAD5* and *SMAD2* were overexpressed on day 14 after chronic TCDD exposure. This might contribute to ovarian cancer progression via TGFβ/activin signaling [74]. *FOXO1* is an essential regulator of endothelial cell proliferation and a tumor suppressor in different cancer types, including liver, endometrial adenocarcinoma, breast, and ovarian cancer [71]. Our data (negative correlation between *FOXO1* and miR-27a-3p expression) suggest that in KGN cells, *FOXO1* might be regulated by miR-27a (Figure 14). In addition, *VEGFB* and *VEGFC* are part of two signaling pathways with major roles in tumor angiogenesis and lymphangiogenesis [75]. According to Zhang et al., *VEGFB* and *VEGFC* contain a potential miR-27a-3p binding site [76], and they were overexpressed in KGN cells.

Lastly, Chen et al. reported that the low expression of miR-8063, a tumor suppressor, leads to activation of WNT/β-catenin signaling, which has been closely associated with the malignant behavior of colorectal cancer [77]. Our data suggest that miR-8063 may be associated with cell transformation in granulosa cells and the bioinformatic analysis identified three putative genes (*VEGFC*, *JAK2,* and *FAS*), also overexpressed in exposed KGN cells. Specifically, *JAK2* upregulation and JAK2/STAT3 pathway activation have critical roles in several solid tumors and significantly influence the survival of patients with ovarian cancers [78]. Conversely, *FAS* upregulation generally increases FAS ligand-induced apoptosis; however, granulosa and KGN cells are normally resistant to FAS ligand-induced apoptosis, except when pre-exposed to cytokines that promote *FAS* expression, such as interferon gamma and tumor necrosis factor alpha [79].

Interestingly, two small nucleolar RNAs (snoRNAs), SNORD49A (U49A) and SNORD49B (U49B), attracted our attention, because they were upregulated upon TCDD exposure for 72 h and downregulated on day 14 compared with control cells (Figure 9). Although they are classical C/D box snoRNAs, usually considered as housekeeping genes for the posttranscriptional modification of rRNAs, several studies have indicated that snoRNAs play oncogenic roles, especially in leukemia [80,81,82,83].

Many different approaches can be used to assess TCDD transgenerational effects, including epidemiological, human, and animal studies, as well as in vitro studies. Each of these approaches has its own strengths and limitations. The major limitation of our in vitro study is that cultured cells are removed from their physiological environment [84]. There are no neighboring cells or tissues to interact with and no blood to supply potentially important factors or nutrients [84]. Moreover, it can be difficult to expose cells in a manner that mimics the in vivo exposure [84]. Increasing evidence indicates that non-DNA sequence-based epigenetic information can be inherited across several generations in different organisms (from yeast to plants and humans) [85]. In humans, transgenerational epigenetic effects have been described in which environmental exposures lead to heritable phenotypic changes that pass through the male or/and the female germline, not somatic cells, as KGN. Nevertheless, many cellular pathways are similarly activated in somatic cells. In general, epigenetic modifications precede genetic alterations in cancer development [86]. In cancer and precancerous cells, changes in miRNA expression and DNA methylation patterns lead to GI, imprinting loss, transcription of proto-oncogenes, and genes encoding proteins involved in GI and metastasis, and they generally precede gene mutations [86]. Therefore, our in vitro study may provide a quick way of testing the hypothesis that EDC-induced miRNA deregulation is a key mechanism in the transgenerational effects of EDCs, while following the Three Rs principle to reduce/replace the use of animals in research. Moreover, epigenetic mechanisms can vary greatly among species and have a more limited effect in mammals than in plants and other animal species [86]. Therefore, results obtained from animal experiments cannot always be translated to humans.

In conclusion, this is the first study that analyzed the effects of chronic exposure to TCDD in vitro in human granulosa cells during several cell cycles to mimic a transgenerational effect. This study showed that TCDD exposure effects are associated with a long-term modification of miRNA levels. TCDD-induced miRNA deregulation may be implicated in the transgenerational transmission of TCDD oncogenic effects [87]. As TCDD is considered an important paradigm for understanding the health consequences of human exposure to EDCs, this study supports the hypothesis that EDC exposure may lead to higher multigenerational cancer risk through miRNA dysregulation transmission. Although epigenetic modifications might be involved, the exact molecular mechanisms leading to miRNA modifications in cells never exposed directly to TCDD need to be deciphered.

## 4. Materials and Methods

### 4.1. Chemicals

TCDD (CAS Number 1746-01-6), 50 µg/mL in dimethyl sulfoxide (DMSO), was purchased from LGC (Teddington, Middlesex, UK) and DMSO (CAS Number 67-68-5) from Sigma-Aldrich (St. Louis, MO, USA). The final concentration of DMSO in culture medium was <0.1%, a concentration that in KGN cells [88] does not have any effect on *CYP19* expression in preliminary experiments by Ernst et al. [47].

### 4.2. Cell Line

The KGN cell line [88] was a kind gift from André Pèlegrin (IRCM, Institut de Recherche en Cancérologie de Montpellier, Montpellier, France). KGN cells showed a pattern similar to that of steroidogenesis in human granulosa cells, thus allowing for the analysis of naturally occurring steroidogenesis in human granulosa cells [88]. The Fas-mediated apoptosis of KGN was also observed, which mimicked the physiological regulation of apoptosis in normal human granulosa cells [84]. Based on these findings, this cell line is considered to be a very useful model for understanding the regulation of steroidogenesis, cell growth, and apoptosis in human granulosa cells [88]. Cells were grown in DMEM/F-12 medium containing 10% heat-inactivated fetal bovine serum and 100 IU/mL penicillin/streptomycin (i.e., culture medium). Cells were grown at 37 °C in a humidified atmosphere with 5% CO_2_, and medium was replaced twice per week. Cells were harvested twice per week with 0.5 mg/ml trypsin/0.2 mg/mL EDTA. KGN had a relatively long population doubling time of about 46.4 h [84]. All culture media and supplements were purchased from Life Technologies Inc. (Life Technologies Inc., Paisley, UK). The absence of mycoplasma was tested every four weeks (MycoAlert LT07-318, Lonza, Walkersville, MD, USA). 

### 4.3. Exposure Schedules

KGN cells were cultured in 6-well plates at a density of 250 × 10^3^ cells/mL. At 24 h post-seeding, cells were exposed to vehicle (<0.1% DMSO; control) or TCDD (1, 10, 50, and 100 nM) for 3, 6, 24, 48, and 72 h. To determine the dose–response curve to TCDD in our culture system, 10-fold serial dilutions of TCDD were prepared from the original stock with DMSO. Each experiment included three biological replicates (exposed and control cells). After incubation for 3, 6, 24, 48, and 72 h, TCDD or DMSO was removed and replaced by fresh culture medium. For the subsequent experiments, the lowest dose that could affect cytochrome P450 1a1 (*CYP1A1*) and P450 1b1 (*CYP1B1*) mRNA expression after acute exposure (3, 6, or 24 h) and chronic exposure (48 and 72 h) was selected.

To identify the effects of acute exposure, cells were exposed to TCDD or DMSO for 3 h and then cultured in fresh culture medium for 48 h before miRNA analysis. To determine the effects of chronic exposure, cells were exposed to TCDD or DMSO for 72 h and then cultured for another 9 or 14 days in culture medium (9- and 14-day samples for mRNA expression and miRNA analysis). Three independent experiments were performed. The 72 h, 9-day, and 14-day samples were from cells at the same confluence level because KGN cells were always cultured in 6-well plates at a density of 250 × 10^3^ cells/mL 4 days before analysis.

### 4.4. Cell Counts

Cells were counted using the trypan blue exclusion assay before and after incubation with vehicle (<0.1% DMSO) or TCDD (1, 10, 50, and 100 nM) for 3, 6, 24, 48, and 72 h and also immediately before each cell passage in 6-well plates at a density of 250 × 10^3^ cells/mL. Cell counts were repeated three times in triplicate experiments. Briefly, 10 μL of cell suspension was stained with 0.4% trypan blue (1:1 *v*/*v*) (Thermo Fisher, Waltham, MA, USA) and viable cells were counted with a hemocytometer (Countess 3, Thermo Fisher).

### 4.5. RNA Extraction

The RNeasy Mini Kit (Qiagen, Valencia, CA, USA) was used to extract total RNA from TCDD- or DMSO-treated cells at the different time-points, according to the manufacturer’s instructions. Total RNA purity and quantity were assessed with a NanoDrop™ One/One C spectrophotometer (Thermo Fisher Scientific, Life Technologies). For samples used for microarray experiments, RNA integrity was assessed with an Agilent 2100 Bioanalyzer (Agilent, Palo Alto, CA, USA). All RNA samples were stored at −80 °C before use.

### 4.6. Quantitative RT-PCR

To assess the biomarkers of TCDD exposure, such as *CYP1A1* and *CYP1B1*, the RNA (0.5 μg) of TCDD-exposed and control samples was used for reverse transcription–quantitative polymerase chain reaction (RT-qPCR) according to the manufacturer’s recommendations (Applied Biosystems, Villebon sur Yvette, France). For qPCR, 2 μL (of a 1:5 dilution) of the first-strand DNA was added to a 10 μL reaction mixture containing 0.25 μM of each primer and 5 μL of 2X LightCycler 480 SYBR Green I Master mix (Roche, Mannheim, Germany). DNA was amplified for 45 cycles with an annealing temperature of 63 °C using the Light Cycler 480 detection system (Roche). The sample values were normalized to the phosphoglycerate kinase 1 (*PGK1*) level using the following formula: *E*^ΔCt^_tested primer_/*E*^ΔCt^_PGK1_ (*E* = 10^−1/slope^), ΔCt = Ct control − Ct unknown, where *E* corresponds to the PCR reaction efficiency. The *E* value was obtained by a standard curve that varies in function of the used primers. Each sample was analyzed in triplicate and multiple water blanks were included. The primers used are listed in Appendix A.

### 4.7. Microarray Hybridization and Data Analysis

Affymetrix microarrays were processed at the Montpellier University Hospital Microarray Core Facility, Montpellier, France (http://www.chu-montpellier.fr/fr/chercheurs/plateformes/les-plateformesrecherche/transcriptome/, accessed on 24 April 2023). Total RNA from KGN cells (200 ng) was labeled for hybridization with the Affymetrix miRNA 4.0 Array (Affymetrix, UK). Each sample was processed individually on a GeneChip miRNA array. Scanned GeneChip images were analyzed using the Affymetrix Expression Console 1.4.1 software to obtain the intensity value signal and the absent/present detection call for each probe set using the default analysis settings and global scaling as the first normalization method. Probe intensities were derived using the robust multi-array analysis (RMA) algorithm.

To identify miRNAs related to TCDD exposure, first the miRNome profiles of three TCDD-exposed and three control (DMSO) samples were compared at each time-point. When fold changes were slightly above 2, three TCDD-exposed and three control samples were added. A first selection was carried out using the detection call (present in at least three TCDD-exposed samples). Then, the Affymetrix expression console software (http://www.thermofisher.com; access date 24 April 2023) was used to identify sncRNAs that were differentially expressed between groups using the ANOVA test (fold change, FC < −2 and >2; false discovery rate, FDR < 5%). The list of differentially expressed sncRNAs (Appendix A) was submitted to Ingenuity (http://www.ingenuity.com; access date 26 August 2023) to identify the predicted target genes of the miRNAs related to acute and chronic TCDD exposure as well as to the progeny of cells after chronic exposure. Hierarchical clustering analyses based on the different expression levels were performed with QIAGEN software packages (https://www.qiagen.com; access date 2 September 2023).

The list of differentially expressed sncRNAs was analyzed and the TCDD and control groups were compared at each time-point using the Wilcoxon matched-pairs signed rank test, *t* test, or Mann–Whitney test, to select sncRNAs for RT-qPCR validation.

### 4.8. TaqMan miRNA Assays

Complementary DNA was synthesized from the total RNA of TCDD-exposed and control cells at different time-points using the TaqMan sncRNA-specific primers miR-27a, miR-24-2*, hsa-miR-30b, RNU49 (ref: #4427975, Life Technologies), miR-8063, and miR-1231 (ref: #4440886, Life Technologies), according to the TaqMan MicroRNA Reverse Transcription Kit protocol (ref: 4366597, Applied Biosystems). For reverse transcription, 250 ng of RNA, 0.15 mL (100 mM) dNTPs, 1 mL of 50 U/mL Multiscribe reverse transcriptase enzyme, 1.5 mL of 10 RT buffer, 0.19 mL of 20 U/mL RNase inhibitor, and 3 mL of 5RT primer (TaqMan MicroRNA Reverse Transcription Kit; Applied Biosystems) were used. Reaction mixtures (15 mL) were incubated first at 16 °C for 30 min, then at 42 °C for 30 min, inactivated at 85 °C for 5 min, and then stored at 4 °C. Quantitative PCR was performed on a Roche LightCycler 480. The PCR reaction mixtures (final volume: 10 mL) included 3 mL of RT product, 5 mL 2TaqMan (No AmpErase UNG) Universal PCR Master Mix (ref: 4324018, Life Technologies), and 0.5 ml of primers from the 20 TaqMan MicroRNA Assay working solution (ref: 4427975; Applied Biosystems). Reaction mixtures were incubated in a 384-well plate at 95 °C for 10 min, followed by 50 cycles at 95 °C for 15 s and at 60 °C for 60 s. The sncRNA expression levels were normalized to the expression level of the reference gene miR-191. The relative sncRNA expression was calculated with the following formula: E_tested sncRNA_^ΔCt^/E_housekeeping miRNA_^ΔCt^ (E = 10^−1/slope^), where ΔCt = Ct control − Ct unknown, and E corresponds to the PCR reaction efficiency. The E value was obtained by a standard curve that varies in function of the used primers. Each sample was analyzed in triplicate and multiple water blanks were included. The mature sncRNA sequences of the TaqMan sncRNA-specific primers used are listed in Appendix A.

### 4.9. MiRNA Target Prediction

Targets of miR-1231, miR-24-2-5p, miR 30b-5p, miR 27a-3p, and miR 8063 were predicted using four online databases: TargetScan7, miRTarBase, miRDB, and DIANA-TarBase v7.0. Unique targets came from the intersection of the results of these four databases with the protein–protein interaction networks derived from Ingenuity (http://www.ingenuity.com accessed the 18 September 2023). Only target genes with a context score percentile > 70 were considered.

### 4.10. Expression of miRNA Target Genes

To determine the expression levels of target genes, RNA (0.5 μg) from TCDD-exposed and control samples was used for RT-qPCR according to the previously described protocol. Each sample was analyzed in triplicate and multiple water blanks were included. The primers used are in Appendix A.

### 4.11. Statistical Analyses

Except for the miRNomic data, statistical analyses were performed with the GraphPad InStat 3 software. Data are expressed as the mean ± SEM or SD and differences between groups were considered significant when the Wilcoxon test gave a *p* < 0.05.

## Figures and Tables

**Figure 1 ijms-25-01144-f001:**
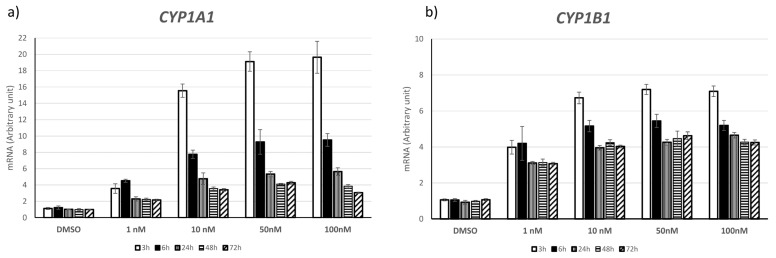
Transcripts of aromatase *CYPA1A* (**a**) and *CYP1B1* (**b**) in KGN cells exposed to different doses and duration of TCDD (arbitrary units with reference to PGK1 housekeeping gene expression in DMSO KGN cells at each duration). The measures are expressed as mean ± SEM (n = 3); *p*-value < 0.05 for all different doses and duration of TCDD vs. DMSO.

**Figure 2 ijms-25-01144-f002:**
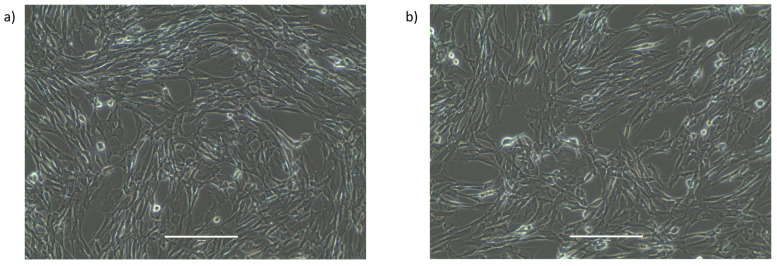
Exemplary images of KGN cells after 72 h exposure. (**a**): untreated (DMSO) cells; (**b**): cells treated with 10 nM of TCDD. Scale bar: 20 μm.

**Figure 3 ijms-25-01144-f003:**
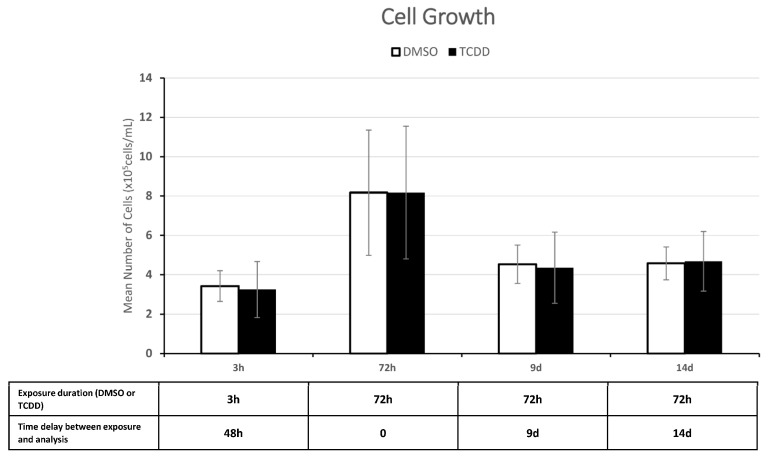
Cell proliferation (trypan blue viability assay) after exposure to DMSO (control) or 10 nM TCDD for 3 h (acute) and 72 h (chronic exposure) and on day 9 and day 14 after chronic exposure end. Data are the mean ± SEM (n = 9); *p*-value > 0.05 for all time-points. Y axis = Mean Number of Cells (×10^5^ cells/mL).

**Figure 4 ijms-25-01144-f004:**
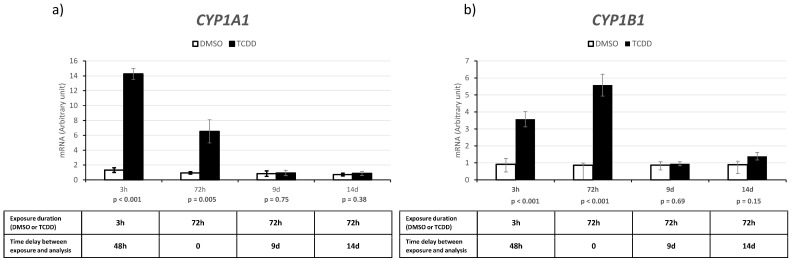
Induction of *CYP1A1* (**a**) and *CYP1B1* (**b**) in KGN cells after acute (3 h) and chronic (72 h) exposure to 10 nM TCDD or DMSO (control) and on day 9 and day 14 after chronic exposure’s end (i.e., cells never directly exposed to TCDD). Data are the mean ± SEM (n = 4) (arbitrary units relative to the expression of the housekeeping gene *PGK1* in control cells). *p* values are indicated for each condition (Student’s *t* test).

**Figure 5 ijms-25-01144-f005:**
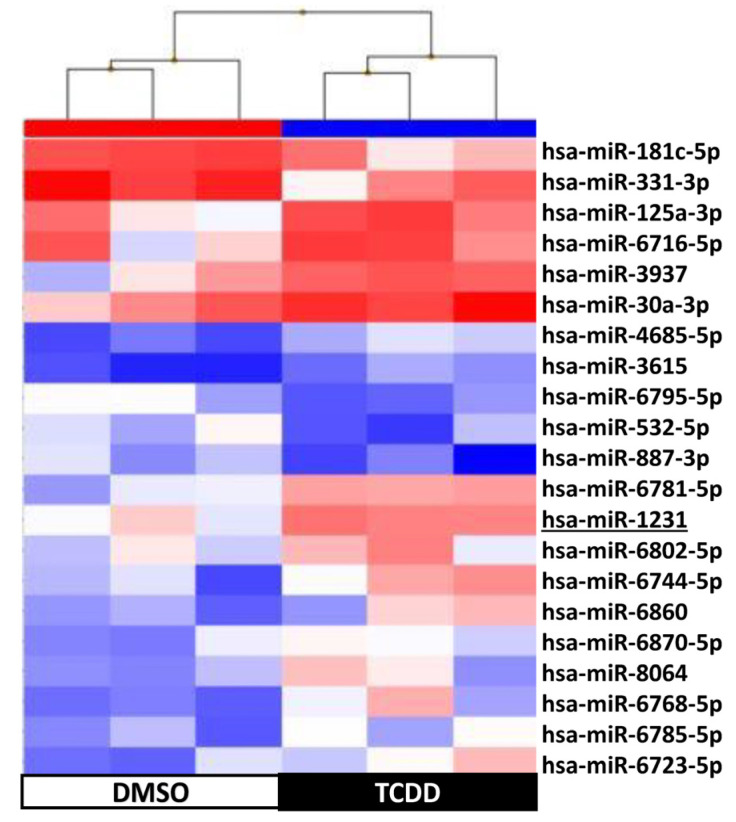
Supervised hierarchical clustering using the 21 small non-coding RNAs differentially expressed in KGN cells after exposure to 10 nM TCDD for 3 h (n = 3; black) vs. DMSO (control, n = 3; white); underlined, miRNAs selected for RT-qPCR validation.

**Figure 6 ijms-25-01144-f006:**
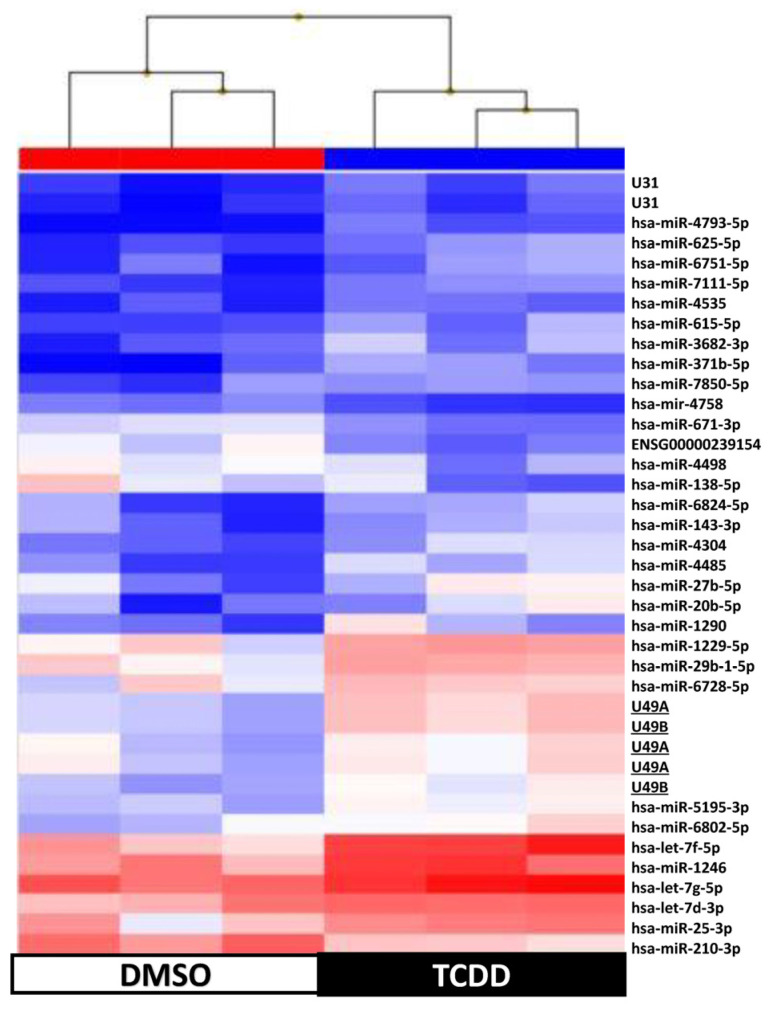
Supervised hierarchical clustering using the 39 small non-coding RNAs differentially expressed in KGN cells after exposure to 10 nM TCDD for 3 h (n = 3; black) vs. DMSO (control, n = 3; white); underlined, miRNAs selected for RT-qPCR validation.

**Figure 7 ijms-25-01144-f007:**
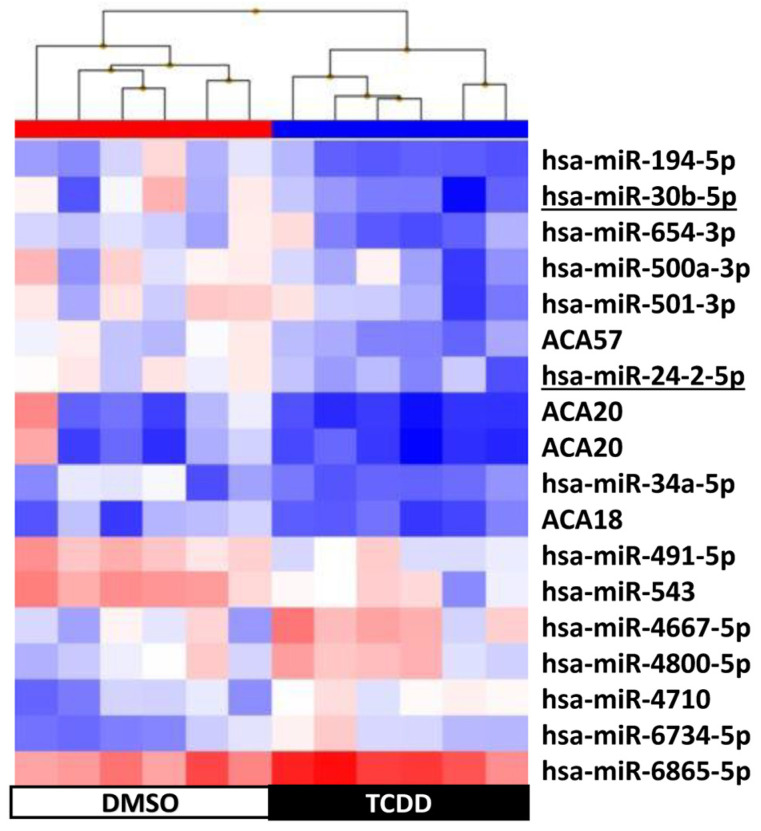
Supervised hierarchical clustering using the 18 small non-coding RNAs differentially expressed in KGN cells on day 9 after exposure to 10 nM TCDD for 3 h (n = 3; black) vs. DMSO (control, n = 3; white); underlined, miRNAs selected for RT-qPCR validation.

**Figure 8 ijms-25-01144-f008:**
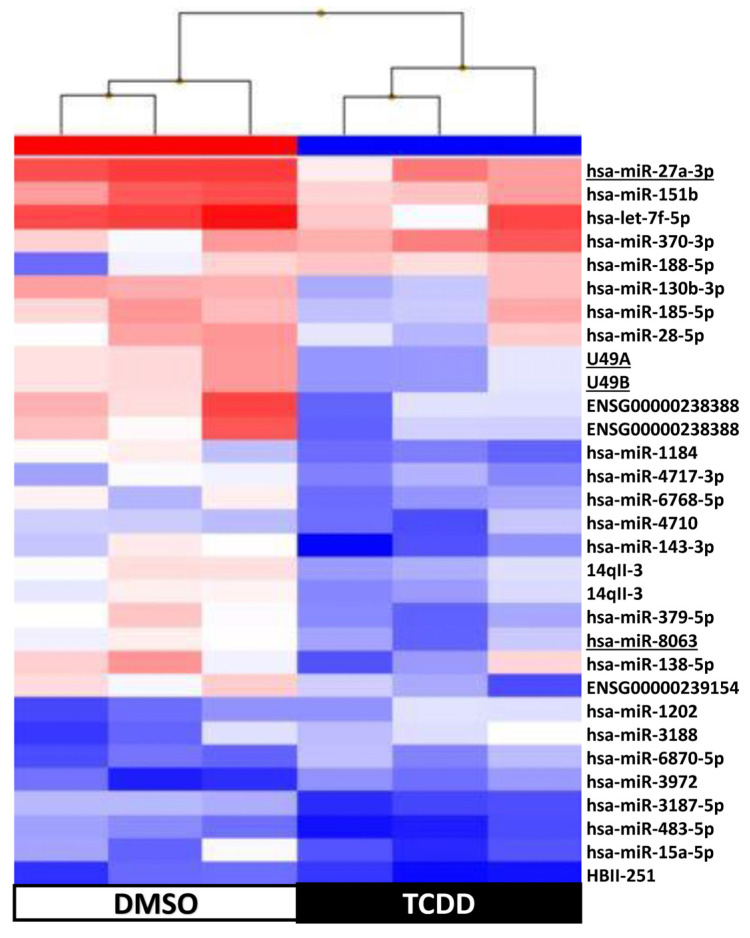
Supervised hierarchical clustering using the 31 small non-coding RNAs differentially expressed in KGN cells on day 14 after exposure to 10 nM TCDD for 3 h (n = 3; black) vs. DMSO (control, n = 3; white); underlined, miRNAs selected for RT-qPCR validation.

**Figure 9 ijms-25-01144-f009:**
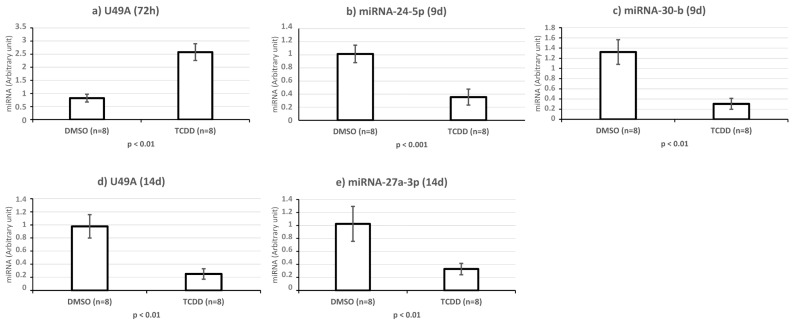
Validation of selected miRNAs in KGN cells (**a**) exposed to 10 nM TCDD for 72 h, (**b**,**c**) on day 9 (9 d) after the end of the 72 h exposure, and (**d**,**e**) on day 14 (14 d) after the end of the 72 h exposure; *p*-values were calculated with Student’s *t* test.

**Figure 10 ijms-25-01144-f010:**
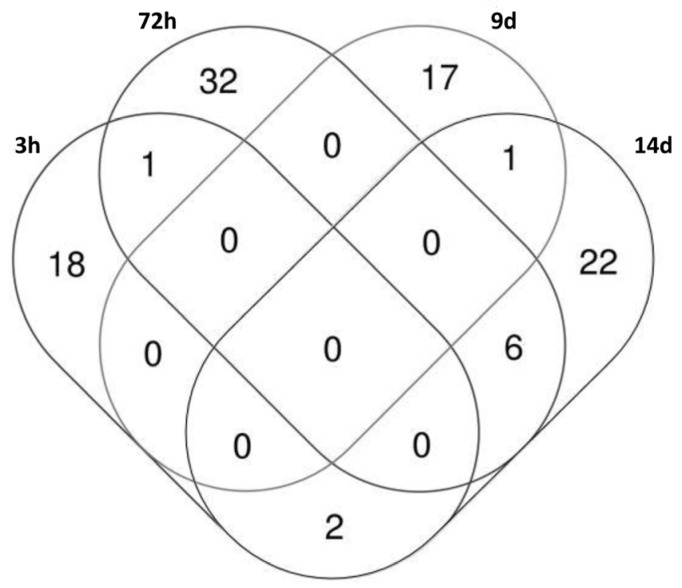
Venn diagram of the 109 small non-coding RNAs differentially expressed at the indicated time-points after exposure to 10 nM TCDD vs. DMSO (control).

**Figure 11 ijms-25-01144-f011:**
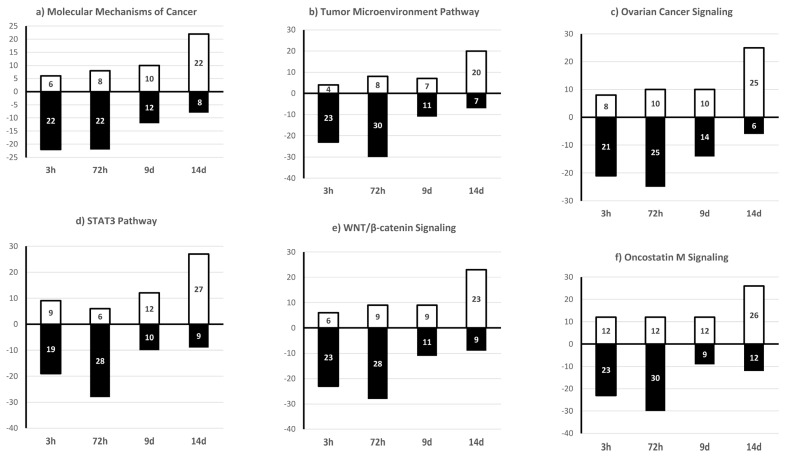
Percentage of transcripts included in each signaling pathway that were predicted by IPA to be upregulated (white) or downregulated (black) after exposure to 10 nM TCDD for 3 h or 72 h and on day 9 (9 d) and day 14 (14 d) after the 72 h exposure end (vs. DMSO, control).

**Figure 12 ijms-25-01144-f012:**
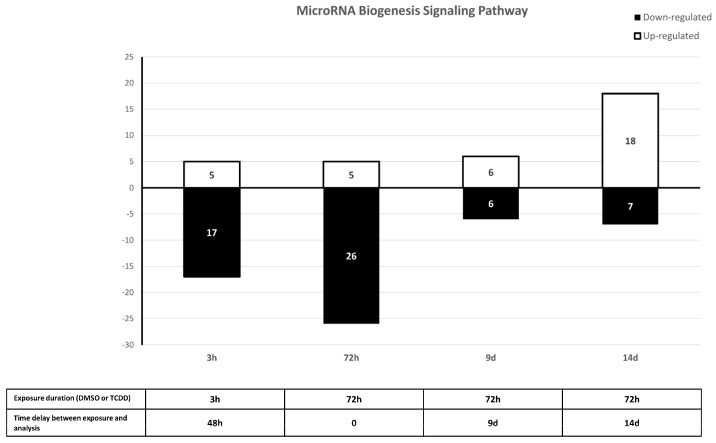
Percentage of transcripts in the MicroRNA Biogenesis Signaling Pathway that were predicted by IPA to be upregulated (white) or downregulated (black) after exposure to 10 nM TCDD for 3 h or 72 h and on day 9 (9 d) and day 14 (14 d) after the 72 h exposure end (vs. DMSO, control).

**Figure 13 ijms-25-01144-f013:**
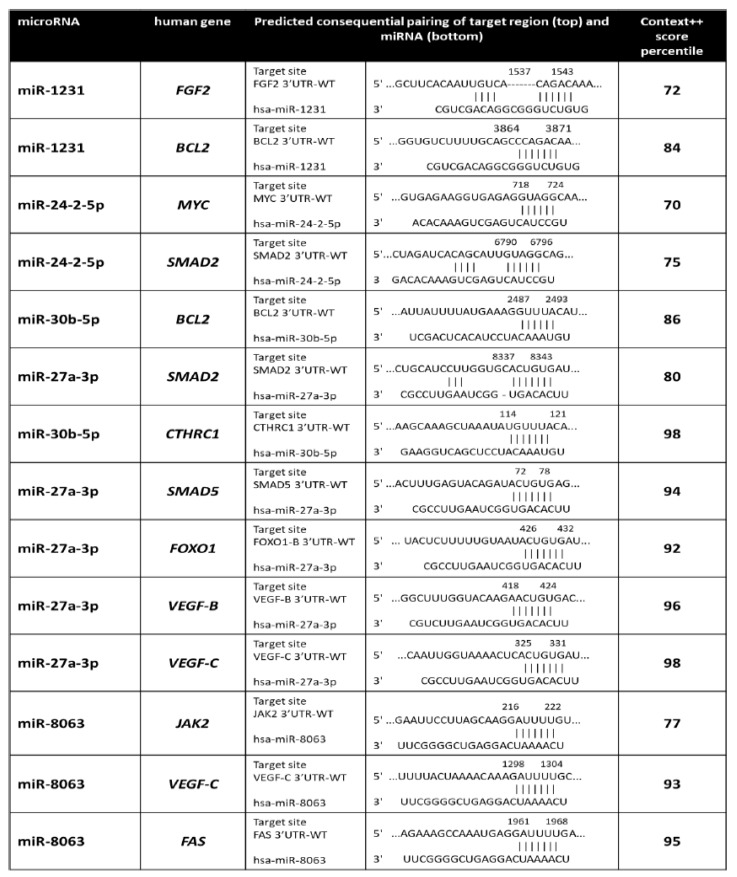
List of the predicted target genes of the indicated miRNAs differentially expressed between TCDD-exposed and DMSO-exposed (control) KGN cells.

**Figure 14 ijms-25-01144-f014:**
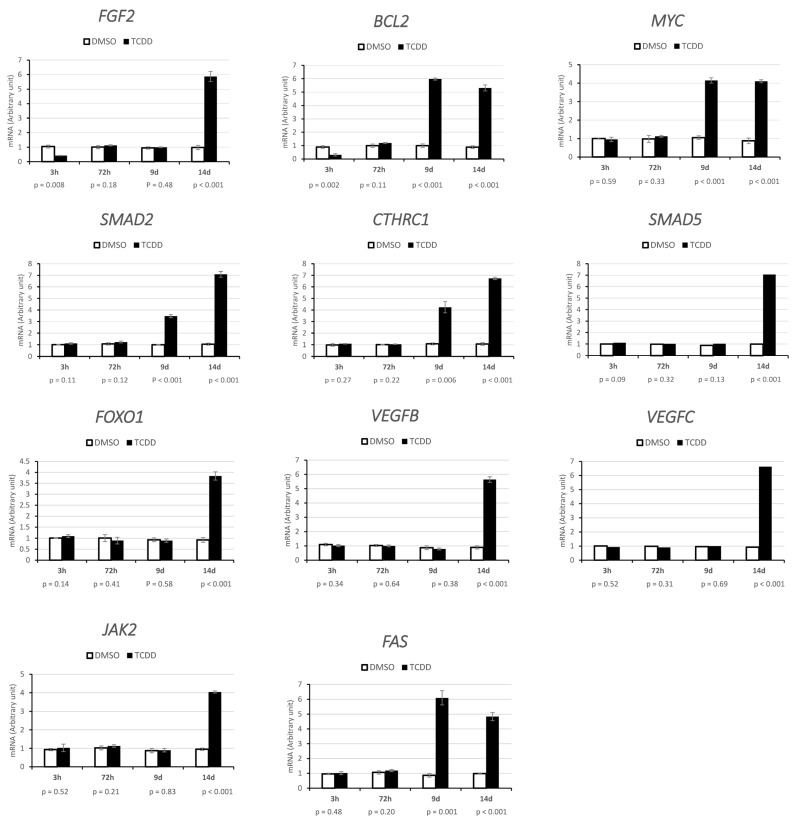
Expression analysis by RT-qPCR of candidate target genes of five miRNAs differentially regulated in KGN cells after acute (3 h) and chronic (72 h) exposure to 10 nM TCDD or DMSO (control) and on day 9 and day 14 after chronic exposure end. Data (arbitrary units relative to the expression of the housekeeping gene *PGK1* in control cells) are the mean ± SEM (n = 3); *p*-values were obtained with Student’s *t* test.

**Figure 15 ijms-25-01144-f015:**
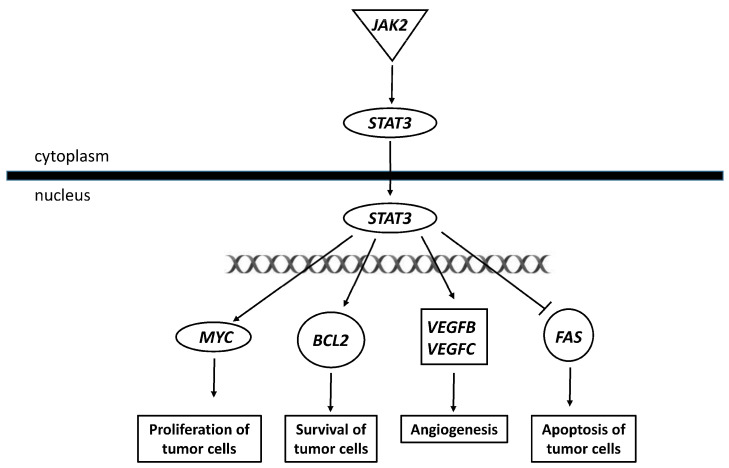
Illustration of the tumor microenvironment pathway generated using QIAGEN’s Ingenuity Pathway Analysis (IPA).

**Table 1 ijms-25-01144-t001:** Number of small non-coding RNAs (sncRNAs) differentially expressed (TCDD vs. DMSO) at each time-point and condition.

	3 h	72 h	9 d	14 d
Exposure duration (DMSO or TCDD)	3 h	72 h	72 h	72 h
Time delay between exposure and analysis	48 h	0	9 d	14 d
Number of differentially expressed snc-RNAs	21	39	18	31

**Table 2 ijms-25-01144-t002:** Top molecular and cellular functions of the 109 small non-coding RNAs (sncRNAs) differentially expressed at each time-point after exposure to 10 nM TCDD vs. DMSO (control). Legend: h = hours; d = days.

Top Molecular and Cellular Function		3 h	72 h	9 d	14 d
	exposure duration (DMSO or TCDD)	**3 h**	**72 h**	**72 h**	**72 h**
	time delay between exposure and analysis	**48 h**	**0**	**9 d**	**14 d**
	number of differentially expressed snc-RNAs	**21**	**39**	**18**	**31**
**Cellular development**		**↓**	**↓**	**=**	**↑**
	number of genes	**1566**	**1774**	**848**	**1584**
	*p*-value range	6.31 × 10^−8^–2.72 × 10^−25^	3.42E × 10^−10^–1.52 × 10^−31^	3.90 × 10^−6^–1.78 × 10^−16^	2.90 × 10^−10^–4.68 × 10^−22^
**Cellular growth and** **proliferation**		**↓**	**↓**	**=**	**↑**
	number of genes	**1472**	**1700**	**772**	**1454**
	*p*-value range	6.31 × 10^−8^–2.72 × 10^−25^	3.42 × 10^−10^–1.52 × 10^−31^	3.90 × 10^−6^–3.74 × 10^−18^	1.42 × 10^−10^–2.37 × 10^−25^
**Cell death and** **Survival**	cell death		**↑**	**↑**	**↓**
	survival		**↓**	**↓**	**↑**
	number of genes		**1644**	**942**	**1433**
	*p*-value range		4.27 × 10^−10^–3.85 × 10^−33^	3.14 × 10^−6^–1.82 × 10^−18^	1.92 × 10^−10^–5.24 × 10^−24^

## Data Availability

All data are in the article or the Appendix A.

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
