# Peer review of "Transgenerational Transmission of 2,3,7,8-Tetrachlorodibenzo-p-dioxin (TCDD) Effects in Human Granulosa Cells: The Role of MicroRNAs"

_ijms, 2024, doi:10.3390/ijms25021144_

Round 1

Reviewer 1 Report

Comments and Suggestions for Authors

The research aimed to investigate the effects of TCDD (2,3,7,8-tetrachlorodibenzo-p-dioxin) on genes involved in aromatase expression, focusing on CYP1A1 and CYP1B1, induced by TCDD exposure. The study utilized KGN cells, resembling human granulosa cells, exposed to varying TCDD concentrations for different durations. The findings revealed significant impacts on CYP1A1 and CYP1B1 expression, particularly at 10nM TCDD concentration over 3 hours. Subsequent experiments used this concentration for both acute (3h) and chronic (72h) exposure. Interestingly, despite the gene expression changes, cell morphology and proliferation remained unaffected. Microarray analysis identified several small non-coding RNAs (sncRNAs) with altered expression upon exposure, with distinct patterns at different time points. Seven specific miRNAs were identified and validated through RT-qPCR, showing varied expression profiles based on exposure duration. These miRNAs exhibited minimal overlap in their deregulation across different time points, highlighting unique molecular responses. Furthermore, Ingenuity Pathway Analysis (IPA) of the differentially expressed sncRNAs suggested their involvement in regulating genes associated with cellular development, growth, proliferation, and cancer-related pathways. Specifically, the study indicated potential downregulation of genes related to cancer pathways after acute and chronic TCDD exposure, followed by their upregulation at day 14 post-exposure. The analysis of predicted targets for these miRNAs indicated potential interactions with genes involved in diverse cellular pathways, including signaling networks associated with cancer promotion, molecular mechanisms of cancer, and cellular transmission. Validation experiments confirmed altered expression levels of specific genes (such as MYC, FAS, BCL2, FGF2, JAK2, VEGF, SMAD5, FOXO1, CTHRC1) at different time points post-exposure, suggesting a dynamic regulatory response to TCDD exposure. Overall, this investigation sheds light on the intricate molecular mechanisms underlying TCDD-induced effects on gene expression and highlights the potential involvement of specific miRNAs in modulating pathways associated with cellular development, proliferation, and cancer-related signaling in KGN cells. 

I have a few concerns that can be addressed:

1.     The title of the article can be grammatically corrected.

2.     sncRNAs, is an umbrella term that includes miRNAs, siRNAs, piRNAs , snoRNAs etc. sncRNAs should not be exclusively used for miRNAs, which is seen in a few places in the text. miRNAs should be introduced early on and also the rationale of picking up specific miRNAs has to be discussed.

3.     In figure 3, lines 117,118 seem to be unedited. Please edit.

4.     In figure 6, the underlined RNAs are not miRNAs but snoRNA which are mostly used as controls. Please include clarification.

5.     Figure 9, the labels on each figure are repeated. Please fix the labels and also the legend corresponding to it.

6.     Figure 11, the labels are missing. Please fix it and the figure legend can be elaborated.

Comments on the Quality of English Language

NA

Author Response

  1. The title of the article can be grammatically corrected. à Thanks, done.
  2. sncRNAs, is an umbrella term that includes miRNAs, siRNAs, piRNAs , snoRNAs etc. sncRNAs should not be exclusively used for miRNAs, which is seen in a few places in the text. miRNAs should be introduced early on and also the rationale of picking up specific miRNAs has to be discussed. à According to your comment, we have modified the mns in this sense.
  3. In figure 3, lines 117,118 seem to be unedited. Please edit. à Thanks, done.
  4. In figure 6, the underlined RNAs are not miRNAs but snoRNA which are mostly used as controls. Please include clarification. à OK, we have added in mns the following sentence « Interestingly, two small nucleolar RNAs (snoRNAs), SNORD49A (U49A) and SNORD49B (U49B), attracted our attention, since they were upregulated upon TCDD exposure for 72h and downregulated at day 14d compared with control cells (Fig. 9). Although they are classical C/D box snoRNAs, usually considered as housekeeping genes for the posttranscriptional modification of rRNAs, several studies have indicated that snoRNAs play oncogenic roles, especially in leukemia [80-83]. »
  5. Figure 9, the labels on each figure are repeated. Please fix the labels and also the legend corresponding to it. à Thanks, we have modified the labels and legends.
  6. Figure 11, the labels are missing. Please fix it and the figure legend can be elaborated. à Thanks, we have modified the labels and legend.

Reviewer 2 Report

Comments and Suggestions for Authors

1. Page 4, line 115 – 10nM, not 10nm.

2. Page 3, line 110 – please include this data in the Supplementary.

3. The graphic side of data presentation requires improvement; The resolution of graphic data needs to be improved - it looks as if the tables were a graphic file - this needs to be changed; the resolution of the charts also needs to be improved; data containing clustering - you can reduce these figures size and improve their resolution; consider introducing colors, e.g. in Fig 1; please zoom out Fig. 10; the signatures and fonts in Fig. 13 require enlargement, if there is a continuation of Fig. 13 on page 14, both parts should be combined into one figure; gene names in graphs, tables and all graphical data should be written in italics.

4. Please add more detailed information about the cell line in the M&M part.

5. Latin phrases such as in vitro etc. in the text they should be italicized.

Author Response

  1. Page 4, line 115 – 10nM, not 10nm. à Thanks, we have modified the mns.
  2. Page 3, line 110 – please include this data in the Supplementary. à Thanks, we have added data.
  3. The graphic side of data presentation requires improvement; The resolution of graphic data needs to be improved - it looks as if the tables were a graphic file - this needs to be changed; the resolution of the charts also needs to be improved; data containing clustering - you can reduce these figures size and improve their resolution; consider introducing colors, e.g. in Fig 1; please zoom out Fig. 10; the signatures and fonts in Fig. 13 require enlargement, if there is a continuation of Fig. 13 on page 14, both parts should be combined into one figure; gene names in graphs, tables and all graphical data should be written in italics. à Thanks, we agree with you! The resolution of the charts are of good quality, but the pagination of document needs to be reviewed, since whichever resolution will seem to be insufficient if tables are enlarged in this way. I hope the journal will do the job. We have modified gene names in graphs and legends.
  4. Please add more detailed information about the cell line in the M&M part. à Thanks, we agree with reviewer’s comment and added in mns “KGN cells showed a pattern similar to that of steroidogenesis in human granulosa cells, thus allowing analysis of naturally occurring steroidogenesis in human granulosa cells [84]. Fas-mediated apoptosis of KGN was also observed, which mimicked the physiological regulation of apoptosis in normal human granulosa cells [84]. Based on these findings, this cell line is considered to be a very useful model for understanding the regulation of steroidogenesis, cell growth, and apoptosis in human granulosa cells [84].”
  5. Latin phrases such as in vitro etc. in the text they should be italicized. à Thanks, done.

Round 2

Reviewer 1 Report

Comments and Suggestions for Authors

1. Figure 3 is completely missing. 

2. Figure 9- still hasn't been changed for labels. There are two b) and two c)'s which is confusing.

3. Figure 11 hasn't been modified yet with no label for the third figure (c?).

Comments on the Quality of English Language

NA

Author Response

I've already modified figures and tables included in this file sent yesteday. I don't know why journal editor has not send you this file and I hope they will use it to modify the text accordingly.

Best regards,

Laura Gaspari
